# Seasonal Stability and Dynamics of DNA Methylation in Plants in a Natural Environment

**DOI:** 10.3390/genes10070544

**Published:** 2019-07-17

**Authors:** Tasuku Ito, Haruki Nishio, Yoshiaki Tarutani, Naoko Emura, Mie N. Honjo, Atsushi Toyoda, Asao Fujiyama, Tetsuji Kakutani, Hiroshi Kudoh

**Affiliations:** 1Center for Ecological Research, Kyoto University, Otsu, Shiga 520-2113, Japan; 2Department of Chromosome Science, National Institute of Genetics, Mishima, Shizuoka 411-8540, Japan; 3Department of Environmental Sciences and Technology, Faculty of Agriculture, Kagoshima University, Korimoto 1-21-24, Kagoshima 890-0065, Japan; 4Department of Genomics and Evolutionary Biology, National Institute of Genetics, Mishima, Shizuoka 411-8540, Japan; 5Advanced Genomics Center, National Institute of Genetics, Mishima, Shizuoka 411-8540, Japan; 6Department of Biological Sciences, Graduate School of Science, The University of Tokyo, Hongo, Bunkyo-ku, Tokyo 113-0033, Japan

**Keywords:** *Arabidopsis halleri*, DNA methylation, natural environment, seasonal changes, seasonally methylated cytosines

## Abstract

DNA methylation has been considered a stable epigenetic mark but may respond to fluctuating environments. However, it is unclear how they behave in natural environments. Here, we analyzed seasonal patterns of genome-wide DNA methylation in a single clone from a natural population of the perennial *Arabidopsis halleri*. The genome-wide pattern of DNA methylation was primarily stable, and most of the repetitive regions were methylated across the year. Although the proportion was small, we detected seasonally methylated cytosines (SeMCs) in the genome. SeMCs in the CHH context were detected predominantly at repetitive sequences in intergenic regions. In contrast, gene-body CG methylation (gbM) itself was generally stable across seasons, but the levels of gbM were positively associated with seasonal stability of RNA expression of the genes. These results suggest the existence of two distinct aspects of DNA methylation in natural environments: sources of epigenetic variation and epigenetic marks for stable gene expression.

## 1. Introduction

DNA methylation at cytosine residues is an epigenetic mark that can be maintained through cell divisions in a wide range of eukaryotic genomes [1]. The previous analyses in diverse organisms have revealed genomic distributions of DNA methylation vary among organisms [2,3,4,5,6,7]. In plants, DNA-methylation varies both between and within species [8], and sometimes is associated with phenotypic variation [9,10,11]. Although the level and patterns of DNA methylation are heritable to a certain extent, the mechanisms that produce and maintain epigenetic variation across generations are largely unknown.

DNA methylation can vary between individuals also by non-genetic causes. It has been shown that both biotic and abiotic treatment can modify DNA methylation [12,13,14]. Because of its semi-stable and semi-labile nature, non-genetic changes in DNA methylation are not explained by simple environmental effects. For example, even in a genetically homogeneous background under stable laboratory conditions, epigenetic variation in DNA methylation can occur during repeated self-pollination in a transgenerational manner [15,16]. Therefore, it is difficult to predict whether DNA methylation is stable or dynamic under natural conditions. Recently the need for “in natura” studies has been highlighted in order to understand how organisms respond to environmental signals by filtering out innumerable fluctuations and noise [17,18]. In the temperate regions, seasonality is the most prominent cause of environmental fluctuations. However, we still do not understand how DNA methylation behaves across seasons.

In order to reveal the seasonal dynamics of DNA methylation, here, we conducted an “in natura” study on genome-wide DNA methylation using a single clone growing in a population of *Arabidopsis halleri* (L.) O’Kane & Al-Shehbaz subsp. *gemmifera* (Matsum.) O’Kane & Al-Shehbaz (hereafter referred to as *A. halleri*), which is a close relative of *Arabidopsis thaliana*. Its clonal propagation and perennial life-cycle allowed us to sample leaves all year round from a single clonal individual [19]. We studied seasonal dynamics of key flowering-time genes and whole transcriptome previously in the site [20,21]. In this study, to understand the dynamics of DNA methylation, we performed whole-genome bisulfite sequencing (WGBS) at 1.5-month intervals, over a year, under natural conditions. We adopted the strategy of monitoring the seasonal pattern in a single clonal individual with a uniform genetic background.

DNA methylation occurs in three contexts, according to the flanking sequence, i.e., CG, CHG, and CHH (H = A, C, or T). The former two form symmetrically and the latter one asymmetrically, in terms of sequences on the complementary strands. Since these contexts of methylation are distinctly regulated and associated with DNA replication, histone modification, and non-coding RNA production [22,23,24], we examined seasonal patterns of DNA methylation by conducting single-base resolution analyses.

The results suggested the existence of seasonally methylated cytosines (SeMCs) in the genome, at least for the examined clone. Interestingly, DNA methylation changes occurred in a context-dependent manner. There were distinct patterns of seasonal changes among CG, CHG, and CHH methylation. Moreover, our analysis revealed that genic CG methylation, i.e., gene-body methylation (gbM), was seasonally stable by itself, and was associated with seasonal stability of RNA expression. This study suggested not only that there is a dynamic nature in DNA methylation in plants in their natural habitats, but also highlights the associations of DNA methylation in robust maintenance of stable RNA expression.

## 2. Materials and Methods

### 2.1. Plant Materials

This study was conducted in a natural population of *Arabidopsis halleri* subsp. *gemmifera* located in central Japan (Omoide-gawa site, Naka-ku, Taka-cho, Hyogo Pref., 35°06′ N, 134°55′ E, altitude 190–230 m). Details of the study site have been described previously [19,20]. Leaf samples were collected at noon on the following dates: 11 November and 22 December 2014, and 9 February, 24 March, 7 May, 23 June, 28 July, and 8 September 2015. In the study site, *A. halleri* forms patches of rosettes that consist of clonally propagated plants and sometimes genetically-related seed-originated plants. Originally six small patches of plants were chosen for leaf sampling, three of them were used for further analyses because the others were heavily damaged by deer herbivory between 11 November and 22 December 2014. At each sampling date, we harvested multiple mature and intact leaves from each plant. Each leaf was ca. 3 cm long, and weighed ca. 0.1 g. To detect DNA methylation and RNA expression in the same set of leaves, a small piece was collected from each leaf for RNA extraction. For DNA extraction, leaves were frozen in an ethanol bath with dry ice, then stored at −80 °C. For RNA extraction, the small pieces of leaves were stored in RNAlater solution (Invitrogen, Carlsbad, CA, USA), then stored at −20 °C according to the manufacturer’s instructions. The data for one patch were analyzed and shown in the main text and figures because the other two patches were found to be genetically mixed (Appendix A). The data for the other two patches were shown in the Appendix A. The samples analyzed here were confirmed to share whole-genome SNPs at levels that were safely judged to be a single clone (Appendix A).

### 2.2. DNA Extraction and WGBS Library Preparation

Genomic DNA was extracted from collected leaves (two leaves per plant, ca. 0.2 g) using the CTAB method [25]. Libraries for WGBS were constructed as described previously [26]. Sequencing was performed with the Illumina Hiseq 2500 system (Illumina, Inc., San Diego, CA, USA).

### 2.3. Processing of WGBS Data

Sequenced reads were trimmed using Trimmomatic [27] (parameters: LEADING:15 TRAILING:15 SLIDINGWINDOW:4:15 MINLEN:36). Trimmed reads were mapped onto the genome sequence of *A. halleri* [28] using Bismark [29] and Bowtie2 [30] with default parameters. Repetitive sequences in the genome were detected using RepeatModeler [31], and repeats with at least 50 bp length were used for further analyses. The level of DNA methylation was calculated for each context using the ratio of the number of methylated cytosines to the number of total sequenced cytosine included in any region of the genome. Efficient bisulfite treatment (>99% in all samples) was confirmed using the level of DNA methylation in unmethylated lambda DNA (Appendix A). SeMC was defined as any cytosine differently methylated between at least two-time points, detected by Fisher’s exact test, with genome-wide FDRs that were calculated using Storey’s method [32]. To draw the heatmaps of methylation of SeMCs, cluster 3.0 [33] and Java Treeview [34] were used. To draw a circos plot for scaffold-wide DNA methylation, Circos software [35] was used. To make browser views of DNA methylation, Integrated Genome Viewer [36] was used. To draw a dendrogram of collected samples, MethylExtract [37], VCF-kit (https://vcf-kit.readthedocs.io), and Dendroscope3 [38] were used with default parameters.

### 2.4. Transcriptome Analysis

RNA extraction and library preparation for RNA-seq were performed using the shotgun type method of BrAD-seq protocol [39]. Sequencing was performed with the Illumina Hiseq 2500 system. Mapping of the reads and calculation of RPM were processed as described previously [21] except that the reference sequence was replaced with a newer annotation [28]. To calculate seasonal average and range of expression of genes more precisely, weekly sampled RNA-seq data from a two-year period [21] were re-analyzed. To quantify the expression, Kallisto software was used [40]. Calculation of seasonal average and range of RNA expression was based on previously described methods [21].

### 2.5. Data Availability

Raw WGBS and RNA-Seq reads are available under the DDBJ BioProject PRJDB7785.

## 3. Results

### 3.1. Seasonal Stability in Large-Scale Distribution of DNA Methylation

To investigate the dynamics of DNA methylation in a natural environment, we analyzed genome-wide DNA methylation in a natural population of *A. halleri* (Figure 1A,B). In the study site, the hourly air temperature ranged from −4.3–36.3 °C during the one-year study period, from November 2014 to September 2015 (Figure 1C). We collected leaves from a single clone of *A. halleri* at 8 sampling times, at 1.5-month intervals across a year, and performed WGBS (Figure 1C). From this, we obtained a series of genome-wide DNA methylation data (Appendix A).

Bulk DNA methylation levels were relatively constant across the eight time points, and ca. 45%, 20%, and 6% of cytosines in the genome were kept methylated in CG, CHG, and CHH contexts, respectively (Appendix A). The large-scale distribution of DNA methylation was determined by the positions in the genome, as represented by radial patterns in the circos plot for the longest 30 scaffolds (30.8% of the reference genome; Figure 2A). The distribution of methylated sites remained constant across the 8 sampling times (represented by 8 concentric circles for each methylation context in Figure 2A). We observed conspicuous aggregation of DNA methylation on repetitive sequences (Figure 2A). For example, scaffolds 3 and 18 are characterized by a low and high density of repetitive sequences that corresponded to relatively low and high methylation levels, respectively (Figure 2A). In the comparative analysis between genic and repetitive regions for the whole genome, the level of DNA methylation in repetitive sequences was higher than that in genes (Appendix A). A similar pattern was confirmed by an analysis using 100 kbp windows for the whole genome (Figure 2B). The level of DNA methylation was correlated with the density of repeats in each window for all three contexts in all samples (e.g., Figure 2B for November 2014, Pearson’s correlation coefficients were 0.64, 0.68, and 0.70 for CG, CHG, and CHH context, respectively; Appendix A for the other sampling times). Although aggregation of DNA methylation at repetitive sequences was similar to the patterns previously reported in related *Brassicaceae* [2,3,41], seasonal stability in the large-scale distribution of DNA methylation was reported for the first time here.

### 3.2. SeMCs: Seasonal Dynamics of DNA Methylation

Next, we searched for SeMCs by detecting differently methylated cytosines in the genome across the eight time points (Fisher’s exact test; *p* < 0.001; FDR < 0.2). The majority of cytosines did not show statistical differences in methylation across the year, and the proportion of SeMCs was less than 0.5% of total cytosines (Appendix A). Still, we detected 62,716, 47,140, and 179,385 SeMCs in CG, CHG, and CHH contexts, respectively, for the examined clone (Figure 3A). They showed diverse seasonal patterns in their level of methylation across the year (Figure 3A). Interestingly, each context of DNA methylation showed distinct patterns of seasonal change: the number of SeMCs that peaked in a particular month was the highest in July, March, and September in CG, CHG, and CHH contexts, respectively (Figure 3B). Differences among contexts of SeMCs could reflect the responsiveness of the regulatory mechanisms of DNA methylation to environmental cues.

### 3.3. CHH SeMCs in Repetitive Sequences

The distribution of SeMCs in the genome differed between the contexts of DNA methylation. The majority of SeMCs in CHH context (CHH-SeMCs) were found in intergenic regions (those annotated as neither exons nor introns) while CG-SeMCs and CHG-SeMCs were in both genic and intergenic regions (Figure 4A). Given that large fractions of the eukaryotic genome are intergenic regions and consist of repetitive sequences such as transposable elements [42,43], we compared the level of CHH methylation for all repetitive sequences in the genome of *A. halleri*. The median level of methylation was the highest in autumn, i.e., September and November (Figure 4B), and the pattern was similar to that of whole-genome CHH-SeMCs. An example of seasonal changes in CHH methylation level in a LINE/L1 showed low and high levels in February–March and September–November, respectively (Figure 4C). The highest methylation levels in autumn were detected in other selected families of repetitive sequences, such as LTR/Copia, LTR/Gypsy, LINE/L1, DNA/MULE-MuDR, DNA/hAT, and RC/Helitron (Appendix A).

In addition, we found some repetitive sequences that showed a unique seasonal pattern of CHH methylation. For example, in the intronic repeat at the locus of *FLOWERING LOCUS C* homolog (*AhgFLC*). *AhgFLC* expression is regulated seasonally and suppresses flowering when it is upregulated [20]. We found that a SINE-like repetitive sequence in the first intron of *AhgFLC* showed a seasonal change of CHH methylation. The level was the highest in March and the lowest in June (Figure 4D). Interestingly, this seasonal pattern was opposite to that of the RNA expression of *AhgFLC* (Figure 4E). The presence of interactions between transcription and CHH methylation in the intragenic repeat at *AhgFLC* is expected.

### 3.4. Gene-Body CG Methylation (gbM) and Constant RNA Expression

Previous studies have reported that gbM is localized to active genes in a wide range of eukaryotes [5,6]. In *A. thaliana*, DNA methylation at the gene body is primarily observed in the CG context [2,3,41]. We examined seasonal patterns of gbM in *A. halleri* and found that the level of gbM was constant across seasons, both in medians and quantiles of all genes (Appendix A), and at the individual gene level (Appendix A).

In order to examine the potential role of gbM in gene regulation, we tested whether the level of gbM associated to seasonal patterns of RNA expression using previously published data of a two-year seasonal RNA-seq of *A. hallei* at the same study site [21]. We found that genes with high gbM often have constant levels of RNA expression across the year. One such example was *AhgPP2AA3* (Figure 5A,B), the homolog of *PROTEIN PHOSPHATASE 2A SUBUNIT A3*, a gene that is known as one that is constantly-expressed under various conditions in *A. thaliana* [44]. This is in contrast to the situation of *AhgFLC* locus, a representative of seasonally expressed genes, which lacks CG methylation from the entire locus, except for the repetitive sequence in the first intron (Figure 5C,D). Based on these observations, we hypothesized that gbM would be associated with constant RNA expression across seasons. To test this hypothesis, we calculated the seasonal average and range of RNA expression for each gene, then compared them between genes with different levels of gbM (Figure 5E,F). Genes that were highly methylated in CG context showed relatively high average levels of RNA expression (Figure 5E). The magnitude of seasonal changes in RNA expression of these genes decreased with increasing levels of gbM (Figure 5F). These results are consistent with the hypothesis mentioned above. The genes with the highest methylation level (in “group 5” in Figure 5E,F) were weakly enriched with two GOs of basic functions (Appendix A).

## 4. Discussion

In this study, we examined the seasonal patterns of DNA methylation at CG, CHG, and CHH contexts in *Arabidopsis halleri,* under natural conditions. The advantage of our study is that we captured dynamics DNA methylation at all seasonal phases that plants actually experience under natural conditions. We identified the genomic sites that showed seasonal changes in DNA methylation. Our observations suggest that seasonal factors in the natural habitat could affect DNA methylation differently according to its context and location in the genome. We would like to note here that our findings in *AhgFLC* were supported by the data for the other two genetically mixed patches: the seasonal dynamics of CHH methylation and RNA expression was reproducibly observed (Appendix A) and the seasonal patterns of RNA expression were also confirmed in the other flowering loci (*AhgVIN3* and *AhgFT*, Appendix A). On the other hand, the genetically mixed patches showed strange patterns of genome-wide bulk DNA methylation (Appendix A). This observation would reflect the existence of a genetic effect on DNA methylation. The genetic and non-genetic effects between different clones should be explored in future population-level studies with multiple repeat samples.

CHH methylation showed seasonal changes in diverse repetitive elements, and the level of DNA methylation was high in autumn, and low in winter. In plants, CHH methylation is regulated by two mechanisms: chromomethylase (CMT) methyltransferases and RNA-directed DNA methylation (RdDM). In *A. thaliana*, previous studies reported that natural variation in *CMT2* gene is associated with local adaptation to climate [45], and that the level of CHH methylation in transposable elements was higher in 16 °C environments relative to 10 °C [13]. CMT2 regulates CHH methylation, especially in long, heterochromatic transposons [22,46]. Additionally, it has been suggested that RdDM pathway are less active at lower temperature and involved in heat tolerance [47,48]. These reports suggest that our observation on CHH SeMCs could reflect the temperature-dependence of the regulation of CHH methylation.

The seasonal change detected in CHH methylation at a repetitive element in the first intron of *AhgFLC* was an interesting example showing how genic heterochromatin behaves in genes. We observed that the seasonal pattern of CHH methylation in this particular repetitive element was opposite to that of the expression of *AhgFLC* gene. *FLC* and its homologs in *Brassicaceae* species contain conserved structures in its intron called VRE (vernalization response element), which is involved in responsiveness to cold treatment [49,50]. Besides, genetic variation of *FLC* locus caused by insertion of TEs into the intron is associated with a low level of RNA expression and with early-flowering phenotype [51,52,53]. In *A. thaliana*, epigenetic changes associated with vernalization have been studied extensively. Expression of *FLC* is repressed by long-term cold treatment (vernalization), accompanied by enrichment of tri-methylation of histone H3 lysine 27 (H3K27me3) at the TSS site at the beginning of cold treatment, and then by H3K27me3 accumulation across gene body region after the plants returned to warmer temperatures [54]. On the contrary, the expression of genes is associated with the removal of H3K27me3 from their bodies [55]. These reports and our results suggest that epigenetic regulation of *AhgFLC* gene might be responsible for seasonal removal of CHH methylation in its intronic repeat. The disruption of transcription in genes has been reported to induce ectopic CHH methylation in *A. thaliana* [56].

Large-scale patterns of DNA methylation were constant throughout the year for all three contexts, although there were seasonal changes in DNA methylation at some cytosine sites. Constant levels of CG methylation were observed in the gene body, and, furthermore, we found that genes with high levels of gbM showed seasonal stability in their gene expression. Although the function of gbM is still unclear [57], our results support previous reports, in *A. thaliana*, that gbM associates with genes modestly expressed among different tissues and experimental conditions [6,58,59,60], and imply the importance of gbM under a seasonal environment in a natural habitat.

Currently, we cannot entirely explain the patterns of DNA methylation in SeMCs, particularly in CG or CHG contexts. It is very likely that there are unidentified processes associated with environmental responses of DNA methylation that have not been studied under laboratory conditions. Because our data were obtained under natural conditions characterized by complex environments, other factors were likely to affect DNA methylation dynamics. For example, aging and mechanical wounding such as cutting the leaves might affect DNA methylation during the perennial life-cycle of *A. halleri*. As mentioned above, the question remains of how the epigenetic variations in plants have been established in the natural fluctuating environment [61,62]. Here, we focused on representative environmental effects on DNA methylation in a single clonal individual. In future studies of DNA methylation, we need to evaluate not only genotype effects, but also, so-called “linage effects” that represent past genetic and environmental effects. Furthermore, we should test to what extent seasonal changes in DNA methylation contribute to the variation in phenotypes in natural environments.

## Figures and Tables

**Figure 1 genes-10-00544-f001:**
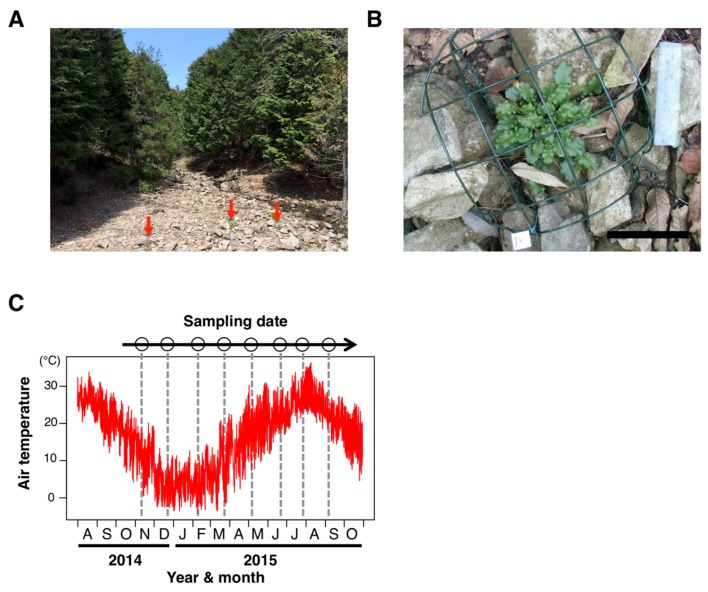
Sampling site and dates for the seasonal DNA-methylation analysis in a natural population of *Arabidopsis halleri* subsp. *gemmifera*. (**A**) A photograph of the study site alongside a small stream through the *Cryptomeria* and *Chamaecyparis* plantation in Hyogo Prefecture, Japan. Red arrows indicate individuals of *A. halleri*. (**B**) An individual of *A. halleri* in the study site. The cage was used for protection against deer herbivory. White bar indicates 100 mm. (**C**) Sampling dates (8 time points) and temperature regimes during the study period. Sampling was performed every ca. 1.5 month for a year. The red line indicates hourly air temperature, and dotted lines represent timings of the sampling.

**Figure 2 genes-10-00544-f002:**
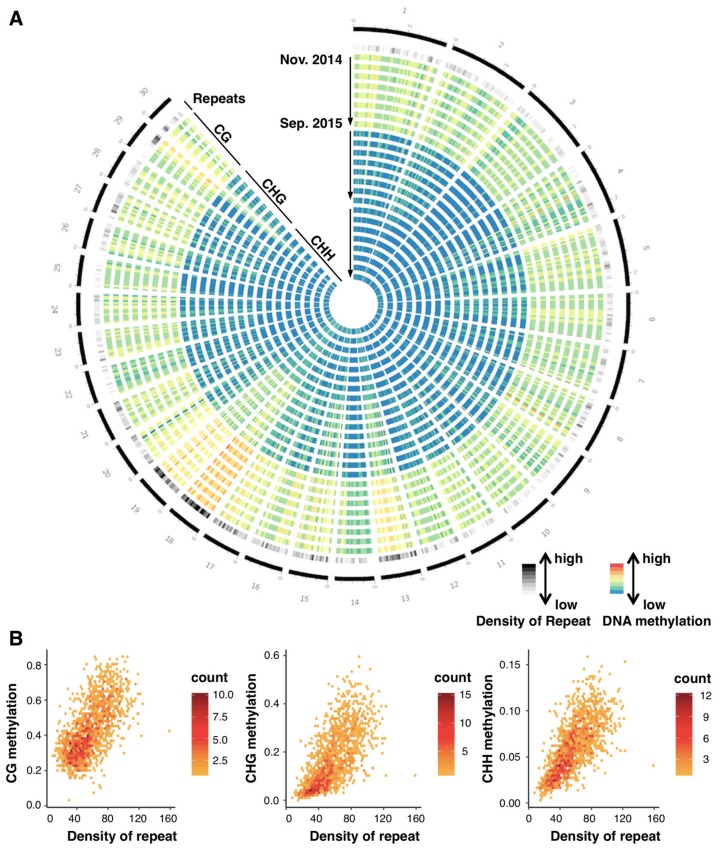
Genomic pattern of DNA methylation across a year. (**A**) A circos plot showing seasonal patterns (8 time points) of DNA methylation in CG, CHG, and CHH contexts for the longest 30 scaffolds of *A. halleri* (30.8% of the genome). On the outermost circle, scaffold positions are indicated by numbers and black bars with scales (one scale = 0.4 Mbp). Each shaded/colored circle represents the scaffold-wide distribution of genomic attributes for each 100 kb window. The second outermost circle represents density of repetitive sequences (including transposable elements). Each tile indicates the density: the lowest in white, the highest in black (0–0.5). The next 24 circles represent DNA methylation levels at 8 time points (starting from 24 November 2014 to 8 September 2015, towards the inner circles shown by the arrows) for CG, CHG, and CHH contexts, respectively. Colors in each tile indicates level of methylation: the lowest as blue, the highest as red (0–0.9 for CG context, 0–0.5 for the others). (**B**) Scattering plots comparing CG, CHG, and CHH DNA methylation against TE density for all 100 kbp windows across the 30 scaffolds.

**Figure 3 genes-10-00544-f003:**
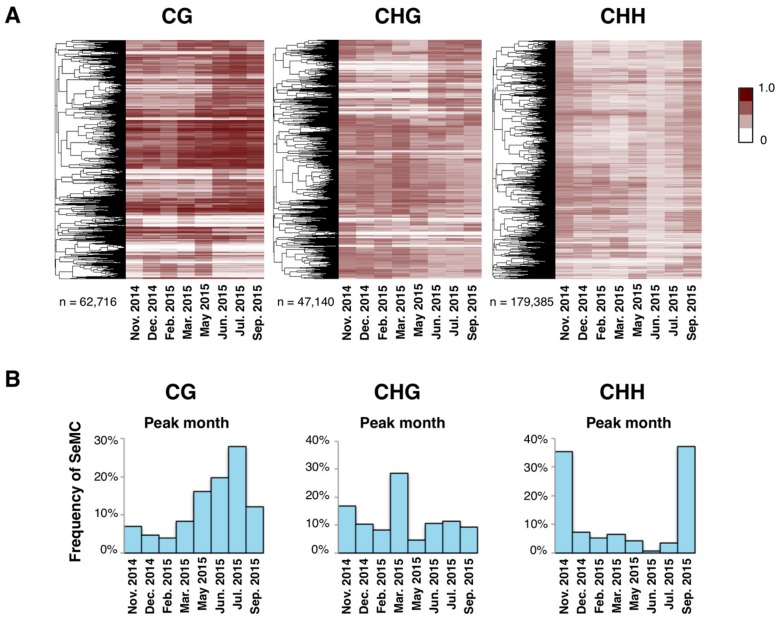
Annual patterns of seasonally-methylated cytosines (SeMCs). (**A**) Heatmaps of seasonally methylated cytosines (SeMCs) in CG, CHG, and CHH contexts (from left to right; n = 62,716; 47,140; 179,385, respectively). Each row indicates a series of DNA methylation ratios across a year in each position in the genome (0: unmethylated, 1; fully methylated). The dendrogram on the left of each heatmap represents the result of hierarchical clustering of SeMCs. Distance in the dendrogram was based on Pearson’s correlation coefficient. (**B**) Barplot showing the distribution of peak timings of methylation level for the SeMCs in CG, CHG, and CHH contexts (from left to right, respectively).

**Figure 4 genes-10-00544-f004:**
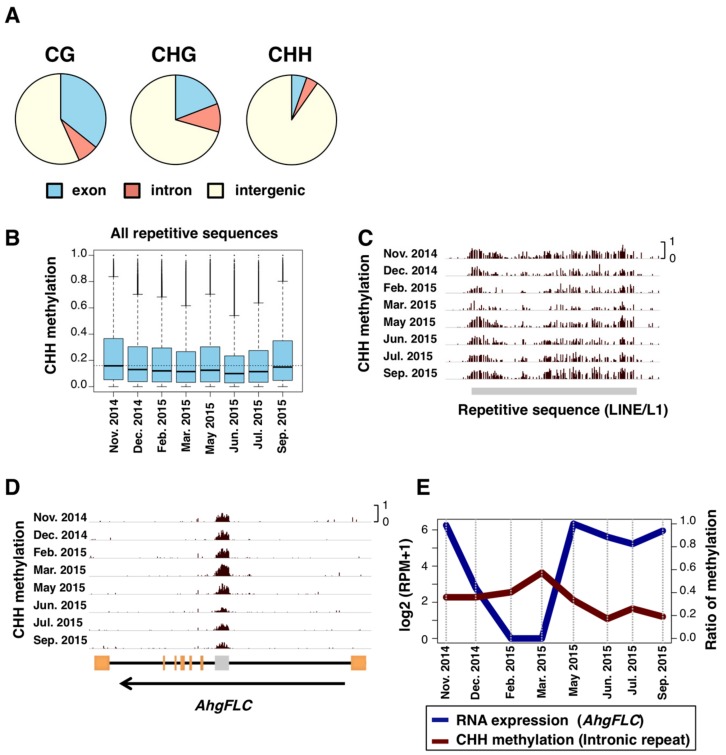
SeMCs in the CG, CHG, and CHH contexts, and seasonal patterns in CHH DNA methylation levels. (**A**) Pie charts indicating locations (exon, intron, and intergenic regions) of seasonally methylated cytosines (SeMCs) in CG, CHG, and CHH contexts (from left to right, respectively). (**B**) Boxplots of CHH methylation at 8 time points in repetitive elements. The boxes span from the first to the third quartiles, the thick black bars inside the boxes are the medians, whiskers above and below the boxes represent 1.5 × interquartile ranges from the quartiles. Dotted line indicates the median CHH methylation level in November 2014. (**C**,**D**) Browser views for seasonal patterns of CHH methylation on repeat sequences in one of LINE/L1 sequences (**C**) and *AhgFLC* locus (**D**). Orange rectangles indicates exons. A gray rectangle in the *AhgFLC* locus indicates a repetitive sequence. (**E**) Comparison between RNA expression of *AhgFLC* and CHH methylation of the repetitive sequence in its intron.

**Figure 5 genes-10-00544-f005:**
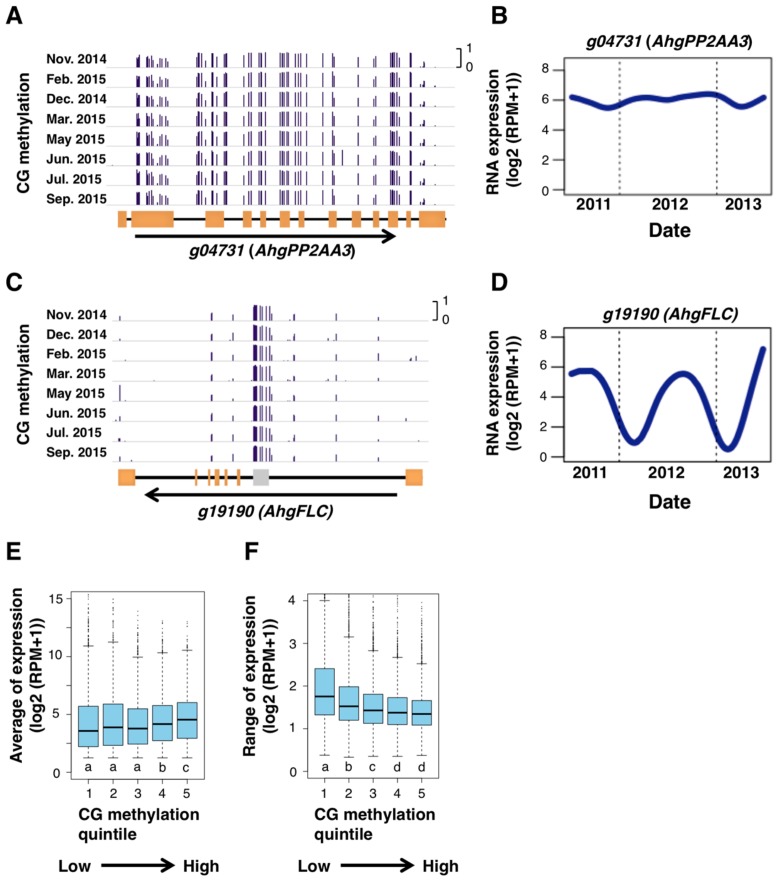
CG DNA methylation and stability of gene expression. (**A**–**D**) Browser views of seasonal patterns of CG methylation (**A**,**C**) and the two-year dynamics of RNA expression level (**B**,**D**; from September 2011 to August 2013) for *AhgPP2AA3* (*g04731*) and *AhgFLC* (*g19190*), respectively. Orange rectangles indicate exons. A gray rectangle indicates a repetitive sequence in the *AhgFLC* locus. (**E**,**F**) Boxplots showing relationship between DNA methylation in CG context and the average (**E**) and range (**F**) of RNA expression of genes. Only expressed genes (average expression level (log2(RPM + 1) > 1)) are used for these analyses. Genes were split into five bins according to quintiles of genic DNA methylation in CG context. The boxes span from the first to the third quartiles, the bands inside the boxes are the medians, whiskers above and below the boxes represent 1.5 × interquartile ranges from the quartiles. Different letters represent significant differences between groups in the Mann-Whitney test, *p* < 0.01 adjusted for multiple comparisons.

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
