# Peer review of "Seasonal Stability and Dynamics of DNA Methylation in Plants in a Natural Environment"

_genes, 2019, doi:10.3390/genes10070544_

Reviewer 1 Report

In the manuscript “Seasonal Stability and Dynamics of DNA Methylation in Plants in a Natural Environment”, the authors analyzed seasonal patterns of genome-wide DNA methylation in a single clone from a natural population of the perennial Arabidopsis halleri. They found that most DNA methylation is stable across the year. They also claimed that the seasonally methylated cytosines (SeMCs) existed.

The data are interesting but my major concern is that it is hard to conclude that the DNA methylation changes are due to the seasons. The authors cannot exclude the possibility that the DNA methylation changes may be caused by aging, mechanical wounding (cutting the leaves) or other factors. So the authors should include these possibilities in the discussion.

minor point:

1, In the methods part, it will be appreciated to provide the parameter detail for the bioinformatics software.

2, Line 131, the authors used the longest 30 scaffolds. It will be appreciated to provide the information that what is the percentage of these longest 30 scaffolds to the whole genome.

3, Line 185 and 186, it seems that "Fig. S4A/B" should be "Fig. S5A/B".

4, Line 191, "Fig 5 A and B", a dot "." is missing after "Fig".

5, For table S2, it will be appreciated to provide more information: 

total cytosine number in the genome

covered cytosine number

covered cytosine number with depth >= 4

average depth per cytosine (using all cytosines)

average depth per covered cytosine (using only covered cytosines)

Author Response

Response to Review #1
Thank you for evaluating our manuscript, and thanks to your comments we are able to improve our manuscript. Our responses to the comments are as follows, and we hope that our revision has made the manuscript ready for publication.

COMMENT: The authors cannot exclude the possibility that the DNA methylation changes may be caused by aging, mechanical wounding (cutting the leaves) or other factors. So the authors should include these possibilities in the discussion.
RESPONSE: Following the comment, we have added discussion about the possibilities suggested by the reviewer to the manuscript (L. 263-266 in the revised version).
COMMENT: 1, In the methods part, it will be appreciated to provide the parameter detail for the bioinformatics software.
RESPONSE: As suggested, we have added information of the parameters in "Materials and Methods" section (L. 97-99, 111 in the revised version).
COMMENT: 2, Line 131, the authors used the longest 30 scaffolds. It will be appreciated to provide the information that what is the percentage of these longest 30 scaffolds to the whole genome.
RESPONSE: The total length of the longest 30 scaffolds is 60,508,839 bp (30.8% of the whole genome). This information has been added to the manuscript (L. 136 and the legend of Fig. 2A in the revised version).
COMMENT: 3, Line 185 and 186, it seems that "Fig. S4A/B" should be "Fig. S5A/B". 4, Line 191, "Fig 5 A and B", a dot "." is missing after "Fig".
RESPONSE: As suggested, we have modified the manuscript accordingly (L. 193-194, 210 in the revised version).
COMMENT: 5, For table S2, it will be appreciated to provide more information
RESPONSE: We have modified Table S2 accordingly by adding the information (Table S2).

Reviewer 2 Report

The manuscript by Ito et al explores seasonal changes in DNA methylation and expression in Arabidopsis halleri through WGBS and RNA-seq. The study concludes that, while overall DNA methylation patterns are stable during the different months tested, there are some changes (SeMCs) over heterochromatic repeats. Moreover, they show that methylation over genes bodies is stable and correlates with stable expression across seasons. The manuscript is well written and the data is well presented.

Previous works have reported a positive correlation between CHH methylation and temperature. The CHH methylation changes reported in this manuscript are not consistent with this observation. Higher CHH methylation is observed in Nov 2014 and Sept 2015, where average temperatures are not higher than temperatures at other time points. Moreover, this methylation pattern does not follow any trend. For instance, if CHH methylation would increase when the temperatures go down (Nov 14 and Sept 15), one would expect to see high CHH methylation in Dec 14 and Jan 15. One concern is the fact that only one replicate from one patch was analyzed, rising doubts about the reproducibility of the methylation results presented. To make this data more reliable authors should show data from different patches. This is especially important regarding the RNA-seq data generated. Considering how noise this technique is, one should do at least 3 biological replicates. No information about replicates is descried in methods. Moreover, if I understood correctly, the RNA-seq data generated in this manuscript is used only to report the FLC expression in Figure 4E? Why did not the authors use the RNA-seq data used for Figure 5?

Considering the changes in heterochromatic CHH methylation reported in this manuscript, it is surprising that the authors do not mention CMT2 in the introduction or discussion. This protein is responsible for most of the heterochromatic CHH methylation and has been shown to be involved in temperature acclimation in Arabidopsis thaliana.

The observation that higher CG methylation correlates with constitutive expression has been reported many times and thus, I do not think these results add much new to the field. Similarly, the regulation of FLC expression by an intronic TE has been well characterized. If reproducible in other patches of plants, the observation that CHH methylation changes over the year and anticorrelates with FLC expression is interesting. It would be interesting to follow the methylation over two years to see if there is a consistent pattern. If so, I wonder if it is an active regulation by methylation pathways (CMT2 or RdDM) or it is a consequence of FLC expression changes.

Line 186. Reference to Figure S4A and S4B should be changed to S5A and S5B

Line 202. Only 2 GOs are shown in Table S4, not 4.

Author Response

Response to Review #2
Thank you for evaluating our manuscript, and thanks to your constructive comments we are able to improve our manuscript. We addressed most of the points raised by this reviewer.

COMMENT: Previous works have reported a positive correlation between CHH methylation and temperature. The CHH methylation changes reported in this manuscript are not consistent with this observation. Higher CHH methylation is observed in Nov 2014 and Sept 2015, where average temperatures are not higher than temperatures at other time points. Moreover, this methylation pattern does not follow any trend. For instance, if CHH methylation would increase when the temperatures go down (Nov 14 and Sept 15), one would expect to see high CHH methylation in Dec 14 and Jan 15.
RESPONSE: That is true that the previous laboratory studies did not explain our observations entirely. We modified the sentences and add description about the discrepancy between our study and the previous studies (L. 260-266 in the revised version). We think that there should be unidentified processes that have not been studied under laboratory conditions.
COMMENT: One concern is the fact that only one replicate from one patch was analyzed, rising doubts about the reproducibility of the methylation results presented. To make this data more reliable authors should show data from different patches. This is especially important regarding the RNA-seq data generated. Considering how noise this technique is, one should do at least 3 biological replicates. No information about replicates is descried in methods.
RESPONSE: As the reviewer suggested, we added the patterns of RNA expression in FLC and two other flowering loci (VIN3 and FT) for other two patches (Fig. S6 in the revised manuscript). Fig. S6 also shows that the CHH methylation pattern in FLC was reproducible among the replicates. These data significantly underpin our observation.
As we describe in the method section, we originally tried to prepare six replicates, but three patches of them were severely damaged by deer herbivory, and the data from two patches (Rep. 2 and Rep.3) were turned out to be mixtures of several genotypes (Fig. S1). Therefore, we focused mainly on Rep.1. A future research using larger size of sampling would reveal the entire picture of whole-genome DNA methylation among the plants in their natural habitat.
Regarding the accuracy and noise, thanks to the reviewer's suggestion, our observations on the dynamics of CHH methylation and RNA expression were underpinned by the data of the other replicates (Fig. S6). Besides, experimental error rates (due to failure of bisulfite treatment and sequencing error) were quite low throughout this study as shown in Table S1. Reflecting the data accuracy, the levels of DNA methylation across the whole genome were very stable across eight time points for most of the genome regions (as shown in Fig. 2A). Although we agree that there is a certain limitation, our data clearly stated their significance in the manuscript.

COMMENT: Moreover, if I understood correctly, the RNA-seq data generated in this manuscript is used only to report the FLC expression in Figure 4E? Why did not the authors use the RNA-seq data used for Figure 5?
RESPONSE: We added the data of RNA-seq for the other two loci (Fig. S6 in the revised version). In the analysis shown in Fig. 5, we used the published data from our previous study (Nagano et al., 2019, Nat. Plants), weekly RNA-seq for two years, because to we needed to estimate the precise range of seasonal RNA expression of each gene. The RNA-seq data generated in this study were obtained at the 1.5 months intervals for a year. Although the data showed clear seasonal dynamics of flowering genes (Fig. S6 in the revised version), they were not frequent enough to estimate the precise range of seasonal RNA expression.
COMMENT: Considering the changes in heterochromatic CHH methylation reported in this manuscript, it is surprising that the authors do not mention CMT2 in the introduction or discussion. This protein is responsible for most of the heterochromatic CHH methylation and has been shown to be involved in temperature acclimation in Arabidopsis thaliana.
RESPONSE: We agree with the reviewer that CMT2 is an important regulatory factor for CHH methylation, especially in long, heterochromatic transposable elements (Zemach et al., 2013, Cell; Stroud et al., 2014, Nat. Struct. Mol. Biol.), and that genetic variation in CMT2 is associated with local adaptation to climate (Shen et al., 2014, PLoS Genet.). Following the comment, we briefly added discussion on CMT2 to "Discussion" section (L. 228-233 in the revised version).
COMMENT: The observation that higher CG methylation correlates with constitutive expression has been reported many times and thus, I do not think these results add much new to the field. Similarly, the regulation of FLC expression by an intronic TE has been well characterized.
RESPONSE: The advantage of our study is that we captured dynamics of gene expression and DNA methylation at all seasonal phases that plants actually undergo under natural conditions. The confirmation of previously reported patterns by "in natura" studies can add a novel strength in this field. We edited the sentences to clarify this point at the beginning of Discussion (L. 217-219 in the revised version). As the reviewer pointed out, previous studies using accessions of A. thaliana showed the existence of intronic TE in FLC locus and its effect on RNA expression and flowering (Gazzani et al., 2003, Plant Phys; Michaels et al., 2003, PNAS; Quadrana, et al., 2016, eLife). Our study offered a new insight into the regulatory system of FLC. We added the description about this point to "Discussion" section (L. 242-244 in the revised version).
COMMENT: If reproducible in other patches of plants, the observation that CHH methylation changes over the year and anticorrelates with FLC expression is interesting. It would be interesting to follow the methylation over two years to see if there is a consistent pattern. If so, I wonder if it is an active regulation by methylation pathways (CMT2 or RdDM) or it is a consequence of FLC expression changes.
RESPONSE: We appreciate the comment greatly. As we described above, the patterns in FLC locus were reproducibly observed in replicates (Fig S6). We agree that the reviewer’s suggestion that analyses for multiple years are very promising, and we are largely encouraged to continue the long-term study on dynamics of epigenetic changes in plants. Although the driving forces of seasonal CHH methylation is unclear, we think that one of those is FLC transcription as described at L. 248-252 in the revised version.
COMMENT: Line 186. Reference to Figure S4A and S4B should be changed to S5A and S5B. Line 202. Only 2 GOs are shown in Table S4, not 4.
OUR RESPONSE: We corrected the points accordingly (L. 193-194 and 210 in the revised version).

Round  2

Reviewer 2 Report

I really appreciate the effort to include methylation data for two other patches. Was this new data generated by WGBS as well? If so, I would definitely show plots similar to Figure 3A and 3B for the two new replicates to confirm the reproducibility of the seasonal pattern of CHH methylation changes described. That would definitely strengthen the conclusions of this work.

Lines 88-89: the fact that two other replicates were analyzed should be described.

Author Response

We would like to thank you for the comments, which help to improve the quality of this manuscript.

COMMENT: I really appreciate the effort to include methylation data for two other patches. Was this new data generated by WGBS as well? If so, I would definitely show plots similar to Figure 3A and 3B for the two new replicates to confirm the reproducibility of the seasonal pattern of CHH methylation changes described. That would definitely strengthen the conclusions of this work.
RESPONSE: The data used in Fig. S6A were generated by WGBS. However, Rep. 2 and Rep. 3 were not suitable for searching SeMCs, because the data for the patches were turned out to be genetically mixed as described in "Materials and Methods" section. In this study, SeMCs were defined as cytosines differently methylated between at least two time points (Fig. 3). So, we could not exclude the possibility that the differently methylated cytosines due to genetic effect were detected as SeMCs. Indeed, there were strange patterns in genome-wide bulk DNA methylation in Rep. 2 and Rep. 3 (Fig. S7): the bulk methylation levels of some samples were oddly high regardless of high efficiency of bisulfite treatment in all samples (Table S1 (modified)). Therefore, we decided to focus mainly on Rep.1 for further analyses. To clarify this point, we modified the sentences and add the information about the levels of bulk DNA methylation in Rep. 2 and Rep. 3. (L. 226-228 in the revised version; Fig. S7).
On the other hand, the pattern of CHH methylation of the intronic repeat in FLC locus exhibited reproducible seasonal pattern of CHH methylation among Rep.1, 2, and 3 (Fig. 4E and Fig. S6). We thus added the data for the other two replicates as supplementary data, in addition to the seasonal expression patterns of the flowering-related genes (FLC, VIN3, and FT; Fig. 4E, and Fig. S6).
The patterns of bulk DNA methylation found in Rep. 2 and Rep. 3 suggested that there would be genetic effect on DNA methylation. A future research using larger size of sampling would explore the genetic effects on DNA methylation.

COMMENT: Lines 88-89: the fact that two other replicates were analyzed should be described.
RESPONSE: As suggested, we have modified the description in "Materials and Methods" section accordingly (L. 88-90 in the revised version).
